# DinoV3–LSTM for Early-Stage Classification of Paroxysmal Atrial Fibrillation in ECGs Toward Prediction

## Abstract

Detecting precursors of paroxysmal atrial fibrillation in long-term ECGs is crucial for timely intervention but remains difficult because precursor signals are subtle and patient physiology varies widely. We present a streamlined framework that adapts a distillation-based vision foundation model, DinoV3, to multi-channel two-dimensional ECG encodings and combines it with an LSTM to form an indicator model for precursor classification. We use three complementary 2D representations—short-time Fourier transform, Gramian angular field, and progressive moving-average transform—to capture time–frequency structure, temporal relationships, and multiscale trends, respectively. Evaluated as a precursor classification study on public long-term ECG datasets, the DinoV3–LSTM pipeline achieves competitive performance despite the absence of ECG-specific fine-tuning. Finally, we outline two concrete next steps to move from precursor classification to clinically useful prediction: ECG-domain fine-tuning with end-to-end optimization, and rigorous lead-time and prospective validation to quantify true prediction capability.

## 1 Introduction

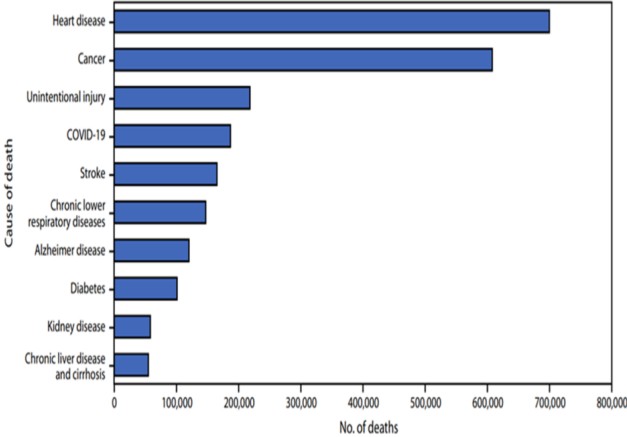

Figure 1: Global mortality statistics related to diseases

Cardiovascular diseases (CVDs) remain the leading cause of global mortality, a figure projected to rise with the accelerating global aging population [1, 2]. Among these, cardiac arrhythmias represent

a significant and often life-threatening subclass. Paroxysmal atrial fibrillation (PAF), a common type of irregular heart rhythm, is particularly dangerous as it can lead to severe complications such as stroke, heart failure, and sudden cardiac death if not detected and managed promptly [3]. While often asymptomatic, PAF can manifest as palpitations, shortness of breath, or dizziness, and its transient nature makes timely diagnosis and intervention challenging. For patients with underlying cardiac conditions or advanced heart failure, the sudden onset of PAF can be critically destabilizing, necessitating proactive and early warning systems.

The advancement of artificial intelligence (AI), particularly in deep learning, has spurred numerous research efforts in ECG-based arrhythmia classification, including PAF detection [4, 5]. These studies have demonstrated impressive accuracy in identifying various arrhythmias from ECG signals. However, a significant limitation of most existing AI-ECG approaches is their focus on classification, determining the current cardiac state based on a contemporaneous ECG recording. While valuable for diagnosis, this retrospective analysis falls short for patients where anticipation of an event is paramount. For vulnerable individuals, such as those with advanced heart failure or other severe cardiac comorbidities, a real-time predictive capability for impending PAF is not merely beneficial but essential for enabling proactive medical intervention and preventing catastrophic outcomes.

In response to this critical need, a growing body of research has explored predictive models for arrhythmias. These studies can generally be categorized into a few types: (1) **Feature-based Predictive Models** that rely on hand-crafted features from heart rate variability (HRV) or P-wave morphology extracted from preceding ECG segments, often utilizing machine learning classifiers like SVM or decision trees [6, 7]. (2) **End-to-End Deep Learning Classification** approaches that use neural networks to directly classify ECG segments into "normal" or "pre-AF" states within a fixed temporal window before onset [8]. (3) More recently, **Self-Supervised Learning (SSL) enhanced models** that leverage unlabeled ECG data to pre-train robust feature extractors. A notable example is a recent study utilizing an MAE-based 1D-ViT [9] for feature extraction, followed by an LSTM network for modeling state transitions and issuing early warnings for both PAF and ventricular fibrillation (VF) [10]. This pipeline demonstrated strong performance.

Despite these advances, the MAE-LSTM pipeline, while effective, presents a key limitation. It typically involves a decoupled training process in which the feature extractor (MAE-based 1D-ViT) and the temporal warning model (LSTM) are trained separately. In many cases, the pre-trained encoder's weights are frozen or only partially adapted during downstream supervised fine-tuning. This separation can restrict task-specific discriminability, as the features learned from a generic masking task may not be optimally tuned for capturing the subtle, evolving patterns of pre-arrhythmic states.

To address this limitation, we propose a novel end-to-end AF early warning framework that replaces the MAE-based feature extraction module with DinoV3, a state-of-the-art self-supervised learning backbone. We adapt DinoV3 to process 2D ECG patch representations (PMAT [11], STFT, GAF) and integrate it seamlessly with an LSTM warning head, enabling joint optimization across the entire pipeline. This end-to-end approach allows DinoV3 to refine its representation learning directly for the early detection of precursor electrophysiological changes, while also benefiting from the strong general priors inherited from its self-supervised pre-training.

It is noteworthy that, in our current study, DinoV3 was employed without additional fine-tuning on large-scale ECG datasets. As a result, its performance is slightly lower than the best-reported MAE-LSTM pipeline. Nonetheless, achieving comparable early warning accuracy without domain-specific fine-tuning underscores DinoV3's potential as a powerful backbone for arrhythmia prediction. We hypothesize that further fine-tuning and fully end-to-end optimization will unlock superior predictive performance. Moreover, DinoV3's attention mechanisms provide interpretable patch-level saliency maps, offering clinicians valuable insights into which ECG segments drive the early warning. This work thus lays the foundation for more accurate, interpretable, and clinically actionable AF early warning systems.

## 2   Related Work

This section reviews existing research pertinent to AF prediction and early warning systems, highlighting the progression from traditional methods to advanced deep learning and foundation models, ultimately setting the stage for our proposed DinoV3-based end-to-end framework.

## 2.1 AF Prediction and Early Warning Research

Early and accurate prediction of AF is paramount for improving patient outcomes and preventing severe complications such as stroke and heart failure [3]. Historically, AF prediction has largely relied on manual feature engineering from electrocardiogram (ECG) signals, combined with conventional machine learning classifiers.

**Traditional ECG Features and Shallow Models.** Classical approaches to AF prediction predominantly utilized hand-crafted features extracted from ECG recordings. These included various heart rate variability (HRV) metrics, which quantify beat-to-beat variations in heart rate, and morphological features such as P-wave duration, QRS complex characteristics, and T-wave abnormalities [6, 7]. Machine learning models like Support Vector Machines (SVMs), Logistic Regression, and XGBoost were then employed to classify these features as indicative of impending AF. While these methods offer a degree of interpretability and have shown promise in certain contexts, they suffer from significant limitations. Their performance is highly sensitive to the accuracy of fiducial point detection (e.g., P-wave onset/offset), which can be challenging in noisy or atypical ECG recordings. Furthermore, these models often require relatively long ECG windows (e.g., 5 minutes or more of R-R intervals) to extract stable and reliable features, which can delay timely warnings [12].

**Deep Learning for AF Classification and Prediction.** The advent of deep learning has revolutionized ECG analysis by enabling automatic feature extraction and classification directly from raw or minimally preprocessed signals [4, 5]. Convolutional Neural Networks (CNNs) have proven highly effective for learning complex spatio-temporal patterns in ECG data, achieving expert-level performance in arrhythmia detection and classification [13]. Recurrent Neural Networks (RNNs), particularly Long Short-Term Memory (LSTM) networks, are well-suited for modeling the sequential nature of ECG data, capturing temporal dependencies crucial for predicting future events [14]. However, many early deep learning methods primarily focused on classification of the current state rather than prediction of future events. While valuable for diagnosis, this approach often falls short in providing the necessary lead time for proactive intervention, especially for patients requiring continuous monitoring for paroxysmal events.

**Pathway Modeling and Early Warning Frameworks.** To overcome the limitations of static classification, more recent research has shifted towards pathway modeling or state-evolution modeling. This paradigm frames the early warning task as monitoring the continuous progression of cardiac states: from a normal state ($S_0$), through a precursor state ($S_1$) characterized by subtle electrophysiological changes, to an eventual event state ($S_2$) like AF onset [15, 16]. Systems built on this concept typically combine powerful feature encoders with recurrent models (like LSTMs) to track these transitions over time. For instance, a recent work successfully utilized a Masked Autoencoder (MAE)-based 1D Vision Transformer (ViT) as a feature extractor, paired with an LSTM for modeling temporal evolution. This pipeline achieved remarkable performance in both PAF and VF early warning tasks, delivering clinically relevant lead times while maintaining high accuracy [15]. This approach optimized a combined metric (e.g., a harmonic metric, HM) that balances accuracy and lead time, demonstrating the feasibility of effective early warning systems. However, a common practice in such SSL-enhanced frameworks is that the pre-trained encoder's weights are often fixed or only lightly fine-tuned during the subsequent warning training phase. This decoupling can potentially hinder the optimal specialization of the feature extractor for the nuanced demands of precursor detection, leaving some discriminative power untapped.

## 2.2 Foundation Models and Self-Supervised Learning for ECG

The concept of foundation models, trained on broad data at scale and adaptable to a wide range of downstream tasks, has gained significant traction across various AI domains [17]. Self-Supervised Learning (SSL) plays a crucial role in developing these models by enabling them to learn powerful, general-purpose representations from unlabeled data, thereby addressing the data scarcity and labeling challenges prevalent in medical fields such as ECG analysis [18].

**Self-Supervised Learning Strategies in ECG.** Common SSL strategies applied to ECG include:

- **Generative Models**: Methods like Masked Autoencoders (MAE) learn to reconstruct masked or corrupted portions of the input signal. This forces the model to capture meaningful underlying data structures [19]. For example, MAE-based pretraining on large ECG corpora using 1D-ViT encoders has demonstrably improved downstream arrhythmia classification performance [9].

- **Contrastive Learning**: Approaches such as SimCLR [20], CPC [21], and BYOL [22] learn representations by maximizing the similarity between different augmented views of the same data sample while minimizing similarity with other samples. These methods have also been successfully applied to learn robust features from ECG data [23].

Despite their effectiveness, a common challenge for these SSL methods, especially when integrated into multi-stage pipelines, is that the pre-trained feature extractors are often treated as fixed components after initial pre-training. This limits their ability to fully adapt to the specific nuances of the downstream task during supervised fine-tuning, potentially restricting their optimal discriminative power for subtle precursor detection.

**Distillation-based Foundation Models.** More recently, distillation-based self-supervised learning, exemplified by models like DINO [24], DINOv2 [25], and its latest iteration, DinoV3[26], has shown immense promise. These models learn rich, dense, and highly transferable features from unlabeled data through a teacher-student knowledge distillation framework. Unlike generative or contrastive approaches, DINO-style models inherently produce robust attention maps and learn powerful representations without relying on explicit masking strategies or complex data augmentations. This paradigm offers a compelling direction for developing general-purpose encoders that can be more seamlessly integrated and adapted within end-to-end learning frameworks, such as for the complex task of ECG-based AF early warning.

## 3 Method

This section details our proposed end-to-end early warning framework for paroxysmal atrial fibrillation (PAF) using a DinoV3-based feature extractor. We outline the overall architecture, describe the datasets utilized for training and evaluation, explain the model's structure.

### 3.1 Overview of the Proposed Framework

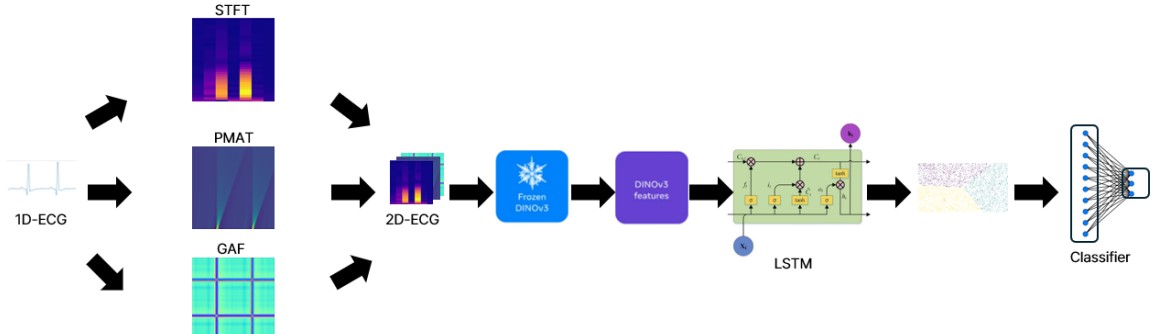

Figure 2: Overall Structure

Our goal is to evaluate whether a vision Foundation Model pre-trained on large-scale image data (DinoV3) can provide robust, transferable representations for ECG state classification without ECG-specific pretraining. The system is organized in two functional blocks for the current study: (1) **Feature extraction** via a DinoV3 backbone adapted to 2D ECG patch representations (PMAT, STFT, GAF), and (2) **Indicator (classification) head** — a temporal model (LSTM) and classifier that maps sequences of extracted features to one of three ECG states (S0: normal, S1: precursor, S2: event). While the overall architecture is designed to support future end-to-end fine-tuning and prospective prediction, here we experimentally isolate and evaluate the classification-stage performance.

## 3.2 Datasets

We used publicly available long-term ECG datasets that are widely used in AF research to ensure reproducibility and comparability.

- **PAF 2001 Dataset**[27]: This database was created for the Computers in Cardiology Challenge 2001 to facilitate the development of automated methods for predicting paroxysmal atrial fibrillation (PAF).It comprises a learning set and a test set, each containing 50 record sets from different subjects. Each record set includes two 30-minute, two-channel ECG recordings. Notably, for subjects with PAF, one 30-minute record immediately precedes a PAF episode, while the other is distant from any PAF activity. This structure is critical for training models to distinguish between pre-PAF states and normal rhythm. The dataset also includes corresponding 5-minute continuation records for verification and QRS annotations.

- **MIT-BIH Normal Sinus Rhythm (NSR) Database**[28]: This database consists of 18 long-term ECG recordings from subjects referred to Boston's Beth Israel Hospital, who were found to have no significant arrhythmias. It includes recordings from 5 men (aged 26 to 45) and 13 women (aged 20 to 50). This dataset provides valuable healthy control data, essential for robustly training the model to recognize and distinguish normal cardiac activity from pre-arrhythmic states.

- **IRIDIA-AF Database**[29]: This dataset comprises 167 Holter monitoring records from 152 patients with paroxysmal AF, collected from an outpatient cardiology clinic in Belgium between 2006 and 2017. Records vary in length from 19 to 95 hours, divided into 24-hour files, and are sampled at 200 Hz. All AF episodes were manually annotated and reviewed by an expert cardiologist and a specialist cardiac nurse. This extensive and expertly annotated dataset offers a rich resource for training and validating models for AF detection and prediction, particularly in real-world long-term monitoring scenarios.

These datasets were preprocessed consistently (normalization, 128Hz resampling, noise reduction filter) to produce training and evaluation sets for the indicator (classification) task.

## 3.3 Model Structure

Our early warning framework is designed with three main components that collaboratively extract discriminative ECG features and capture states.

- **DinoV3-based Feature Extractor**: To enhance representation learning, we replace the original MAE-based 1D Vision Transformer with DinoV3 as the backbone. Raw 1D ECG signals are converted into multi-channel 2D representations using Progressive Moving Average Transform (PMAT), Short-Time Fourier Transform (STFT), and Gramian Angular Field (GAF). These representations are fed into DinoV3, a self-supervised vision transformer pre-trained on large-scale image datasets. DinoV3 provides robust high-level semantic features and interpretable attention maps, enabling the detection of subtle electrophysiological variations related to arrhythmia onset.

- **LSTM-based Classification indicator Model**: An LSTM network processes the sequential state indicators to model the temporal evolution of ECG states. By capturing long-range dependencies, the LSTM effectively learns the progression pathway from stable rhythm (S0), through precursor states (S1), to arrhythmic events (S2). This enables accrately classification for impending arrhythmias.

This Two-state annotation is used solely to train and evaluate the classification (indicator) model in this study; moving from accurate indicator classification to reliable prospective event prediction is left to future work.

## 3.4 ECG State Annotation

Accurate labeling of sequential segments is critical for training the indicator model. We adopt a sliding-window scheme with a 10-second window and 5-second stride; labels are assigned based on the window endpoint relative to the annotated event onset $T_{\text{onset}}$.

- **Normal State (S0)**: segments whose window endpoint lies more than 20 minutes before $T_{\mathrm{onset}}$. These segments are considered to represent stable, baseline rhythm.

- **Precursor State (S1)**: segments whose window endpoint falls within the 20-minute interval immediately preceding $T_{\mathrm{onset}}$. These segments may contain subtle alterations that precede the AF episode.

- **Event State (S2)**: segments whose window endpoint is at or after $T_{\mathrm{onset}}$ and which contain the arrhythmic event.

These states are meticulously annotated to serve as ground truth for training the LSTM-based indicator model, enabling it to recognize and classify the ECG state of individual segments. In the future, the warning model utilizes these classifications to identify the critical transition pathways from S0 to S1 and ultimately to S2.

# 4 Experiments

This section presents the experimental setup and results for evaluating the classification performance of the DinoV3-based indicator model. Our primary goal is to demonstrate the effectiveness of using DinoV3 as the feature extractor.

## 4.1 Evaluation of Indicator Model Classification Performance

We compare the DinoV3-based indicator model against representative baselines, including classical deep ECG classifiers and recent SSL-enhanced pipelines adapted to ECG.

| Model | Acc | Spe | Sen | F1 |
|---|---|---|---|---|
| $\mathrm{FCN}_{wang}$[30] | **0.6124** | **0.7567** | 0.5944 | 0.5673 |
| ConvNeXt[31] | 0.4967 | 0.6667 | 0.3333 | 0.2212 |
| ViT1D[32] | 0.5325 | 0.7169 | 0.3968 | 0.1541 |
| Trs[9] | 0.5900 | 0.7561 | **0.6416** | **0.6208** |
| DinoV3 (Proposed) | 0.5950 | 0.7479 | 0.5368 | 0.5489 |

Table 1: Comparison of classification performance by changing feature extractor

Table 1 illustrates the classification performance for the paroxysmal Atrial fibrillation event (PAF). When trained for classification, the DinoV3-based indicator model achieves competive results across all metrics (Acc, Spe, Sen, MCC, F1) compared to other baselines, including traditional ECG classification models and contrastive learning models. This demonstrates that DinoV3 exhibits enhanced capabilities in extracting discriminative features for ECG state recognition even without the ECG dataset fine tuning, thereby establishing a robust foundation for the subsequent warning model. These results suggest that DinoV3's powerful feature extraction abilities can effectively contribute to detecting various ECG abnormalities.

# 5 Discussion and Limitations

In this study we evaluated the utility of a publicly available Foundation Model (DinoV3) — pretrained on large-scale image data and applied *without* ECG-specific pretraining — as a feature extractor for an ECG-state **classification** (indicator) model targeting paroxysmal atrial fibrillation (PAF). Our principal finding is that DinoV3-derived representations, when applied to 2D ECG patch transforms (PMAT, STFT, GAF) and coupled with a temporal classification head (LSTM), yield competitive indicator-stage performance (Table 1). This result highlights notable cross-domain transferability: visual representations learned at scale retain structure that can be exploited for biological time-series classification, at least at the upstream classification (indicator) level.

Crucially, we reiterate that the present work focuses on the **classification/indicator stage** — i.e., recognizing and distinguishing ECG segments that are normal (S0), precursor (S1), or event (S2). We did not perform prospective time-to-event prediction experiments in this manuscript. Translating accurate indicator classification into reliable early-warning or prediction requires additional modeling

choices (e.g., explicit temporal forecasting, survival analysis, calibration over time) and rigorous temporal validation; these steps are outside the scope of the current paper and are discussed below as future directions.

Despite promising indicator-stage results, there are important limitations:

- **No ECG-specific pretraining in main experiments.** The DinoV3 backbone was used primarily in its publicly released image-pretrained form. While this allowed us to probe cross-domain transferability, we expect that ECG-specific pretraining and/or systematic fine-tuning would further improve discriminative power for precursor patterns.

- **Classification vs. predictive utility.** Strong performance on S0/S1/S2 classification does not automatically imply robust prospective prediction of PAF onset. Predictive performance must be evaluated under temporal holdout schemes that respect event timing, and with metrics appropriate for forecasting (e.g., lead-time distribution, time-dependent AUROC, calibration of predicted risk).

- **Dataset diversity and generalization.** Public long-term ECG datasets with high-quality, temporally precise annotations for precursor intervals are limited. Broader external validation across multiple cohorts, devices, and recording conditions is necessary to assess generalizability and to avoid dataset-specific biases.

- **Clinical validation and interpretability.** Although DinoV3 attention maps provide patch-level saliency that may aid interpretability, these visualizations require formal evaluation by clinicians to determine clinical relevance and to avoid misleading explanations. Human-in-the-loop validation and prospective pilot studies are needed before clinical deployment.

Based on the above, we identify concrete next steps to bridge the gap between the present indicator-stage findings and clinically useful predictive systems:

1. **ECG-specific fine-tuning and ablation studies**: systematically fine-tune DinoV3 on large ECG corpora, compare frozen vs. partially fine-tuned vs. fully end-to-end settings, and report corresponding gains in indicator and (eventually) prediction metrics.

2. **Temporal/predictive modeling**: extend the pipeline to explicit time-to-event or sequence-forecasting models and evaluate under temporally strict validation (e.g., by-record or by-patient splits, prospective simulation).

3. **External and prospective validation**: test on heterogeneous external datasets and, where possible, conduct prospective pilot evaluations to measure real-world utility, false alarm rates, and clinician acceptance.

4. **Clinician-centered interpretability studies**: present attention/saliency maps and decision traces to expert cardiologists for qualitative and quantitative assessment.

5. **Robustness and calibration**: evaluate model calibration over lead time and implement uncertainty-aware decision thresholds suitable for clinical workflows.

Taken together, the current results establish a promising proof-of-concept: Foundation Models can serve as effective upstream feature extractors for the *classification* stage in AF early-warning pipelines, but additional work (fine-tuning, temporal prediction modeling, and clinical validation) is required to realize reliable prospective warning systems.

# 6  Conclusion

We investigated the feasibility of using a vision Foundation Model (DinoV3) as a feature extractor for an ECG-state classification (indicator) model relevant to paroxysmal atrial fibrillation (PAF). By converting ECG signals into 2D patch-based representations and coupling DinoV3 embeddings with a temporal LSTM classifier, we demonstrated competitive indicator-stage performance despite using DinoV3 without ECG-specific pretraining. These findings indicate meaningful cross-domain transferability and position Foundation Models as a viable starting point for building upstream components of AF early-warning systems.

It is important to emphasize the scope of this work: the experiments target the **classification (indicator)** stage that precedes prospective prediction. We do not claim to have developed a validated

predictive early-warning system in this manuscript. Rather, our contribution is to show that (1) Foundation Models can yield useful representations for ECG-state discrimination without domain-specific pretraining, and (2) these representations provide a strong foundation for subsequent work on fine-tuning, end-to-end optimization, and time-to-event prediction.

Future work will prioritize systematic ECG fine-tuning of the backbone, extension to explicit predictive modeling with temporally rigorous evaluation, comprehensive external validation, and clinician-in-the-loop interpretability studies. Addressing these steps is essential to translate the present proof-of-concept into clinically actionable early-warning tools that improve patient outcomes.

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

## Agents4Science AI Involvement Checklist

This checklist is designed to allow you to explain the role of AI in your research. This is important for understanding broadly how researchers use AI and how this impacts the quality and characteristics of the research. **Do not remove the checklist! Papers not including the checklist will be desk rejected.** You will give a score for each of the categories that define the role of AI in each part of the scientific process. The scores are as follows:

- **[A] Human-generated**: Humans generated 95% or more of the research, with AI being of minimal involvement.

- **[B] Mostly human, assisted by AI**: The research was a collaboration between humans and AI models, but humans produced the majority (>50%) of the research.

- **[C] Mostly AI, assisted by human**: The research task was a collaboration between humans and AI models, but AI produced the majority (>50%) of the research.

- **[D] AI-generated**: AI performed over 95% of the research. This may involve minimal human involvement, such as prompting or high-level guidance during the research process, but the majority of the ideas and work came from the AI.

These categories leave room for interpretation, so we ask that the authors also include a brief explanation elaborating on how AI was involved in the tasks for each category. Please keep your explanation to less than 150 words.

IMPORTANT, please:

- **Delete this instruction block, but keep the section heading "Agents4Science AI Involvement Checklist",**

- **Keep the checklist subsection headings, questions/answers and guidelines below.**

- **Do not modify the questions and only use the provided macros for your answers**.

1. **Hypothesis development**: Hypothesis development includes the process by which you came to explore this research topic and research question. This can involve the background research performed by either researchers or by AI. This can also involve whether the idea was proposed by researchers or by AI.

   Answer: **[TODO]** [B]

   Explanation:**[TODO]** AI informed trend analysis, but the research direction and core ideas were proposed by the researcher

2. **Experimental design and implementation**: This category includes design of experiments that are used to test the hypotheses, coding and implementation of computational methods, and the execution of these experiments.

   Answer: **[TODO]** [C]

   Explanation: **[TODO]** Researcher designs experiments; AI handles coding and execution, with researcher refining errors or logic gaps.

3. **Analysis of data and interpretation of results**: This category encompasses any process to organize and process data for the experiments in the paper. It also includes interpretations of the results of the study.

   Answer: **[TODO]** [C]

   Explanation: **[TODO]** Researcher designs experiments; AI handles coding and execution, with researcher refining errors or logic gaps.

4. **Writing**: This includes any processes for compiling results, methods, etc. into the final paper form. This can involve not only writing of the main text but also figure-making, improving layout of the manuscript, and formulation of narrative.

   Answer: **[TODO]** [D]

   Explanation: **[TODO]** AI drafted the overall framework; I manually revised the logical flow and experimental-result interpretations to correct overinterpretation.

5. **Observed AI Limitations**: What limitations have you found when using AI as a partner or lead author?

   Description: **[TODO]** AI often risks overinterpretation in experiments and lacks creativity in topic selection, yet proves useful when merging diverse fields.

